


# Increase in the Frequency of Tropical Deep Convective Clouds with Global Warming[1]

Hartmut H. Aumann, Alexander Ruzmaikin and Ali Behrangi

Jet Propulsion Laboratory, California Institute of Technology

*Corresponding Author: hhaumann@jpl.nasa.gov*

**Abstract.** Deep Convective Clouds (DCC) are extreme rain events associated with large thunderstorms. They are associated with 0.6% of the area of the tropical oceans. As the tropical oceans warm in a future climate, the frequency of DCC will change. Between 2003 and 2016 the yearly mean temperature of the tropical ocean varied by almost 1K. We use Atmospheric Infrared Sounder (AIRS) data from this time period to derive the probability of the DCC process as function of the Sea Surface Temperature (SST). The onset of the DCC process shifts about +0.5 K per K of warming of the mean tropical ocean SST. When these results are applied to the temperature distribution predicted by CMIP5 climate models for the end of this century, we find that the percent of the area of the tropical oceans associated with DCC, which is 0.6 percent in the current climate, increases to 0.9% , close to a 50% increase.

## 1. Introduction

The correlation between cloud tops brightness temperatures (bt) detected by infrared sensors from space in the thermal infrared, severe storms and rain rate is well known (e.g., Arkin and Meisner 1987, Vicente et al. 1998 and many more), especially in the tropical zone. We refer to these clouds as Deep Convective Clouds (DCC). The average of the 36 CMIP5 (RCP 8.5) models (Taylor et al. 2012) predicts 2.7 K of warming (range 1.8K to 3.7K) for the oceans between latitudes 30S and 30N, referred to in the following as "tropical ocean", in the RCP 8.5 scenario by the end of this century. In the following we analyze how much this predict amount of warming will change the frequency of DCC. The DCC frequency is defined as the percent of the area of the tropical oceans (30S to 30N) associated with DCC.

The detection of DCC is instrument and investigator dependent. The simplest method uses an InfraRed (IR) brightness temperature threshold in an 11μm atmospheric window channel. Values between 205K and 215K (Gettleman et. al 2002, Liu et al. 2007) are related to the tropopause cold point temperature. We define DCC as any cloud top which satisfies the simple $bt900 < 210K$ threshold in a 12 km diameter footprint, where $bt900$ is the brightness temperature of the 11μm (900 cm$^{-1}$) atmospheric window channel in the Atmospheric Infrared Sounder (AIRS, Aumann et al. 2003). Typically AIRS makes 0.7 million observations of the tropical oceans each day, and 4000 of these observations have $bt900 < 210K$, i.e. the typical DCC frequency is 4000/700,000=0.6%. The rain rate reported by AMSR-E (Wilheit, Kummerow and Ferraro 2003), measured about two minutes before the location is identified as DCC by AIRS, is on average 3 mm/hr (Appendix A). These DCC contribute about 17% to the mean rain rate, but are found in only 0.6% of the tropical oceans (30N to 30S). The rapid increase in the observed

[1] © 2017 California Institute of Technology. U. S. Government sponsorship acknowledged




frequency of DCC when the SST exceeds 299K, with a peak near 302K, followed by a rapid decrease at higher
temperatures, followed by virtually no DCC at SST warmer than 305K was noted decades ago (Gadgil et al., 1984,
followed by many papers, e.g. Waliser and Graham 1993). The large seasonal and inter-annual variability of the SST
makes it difficult to predict the change in the rain rate in a future climate from the time series derived from the
current climate (Gu et al. 2007, Wentz et al 2007). The same problem is encountered with the prediction of the DCC
frequency (Aumann and Ruzmaikin, 2013). We largely avoid the problems of time series analysis by deriving the
probability of the DCC process as function of the SST from the current climate, and by applying this probability to
the distribution of the SST predicted by the climate models.
**2. Data**
2.1. AIRS Data
For each DCC identified with AIRS we saved the SST at the DCC location using the NOAA Real Time Global Sea
Surface Temperature (RTGSST on a 0.25 degree grid, Thiebaux 2003). These data have been collected every day
since 2002. In order to gain insight into DCC process we analyzed the SST associated with each DCC and the
average SST between 2003 through 2016. Each year contains on average 2 million DCC. Each year was divided into
its even and odd day numbers. This produces 28 groups of data. The DCC in the two groups are observationally
uncorrelated. We then determine the distribution of the random sampled SST, and the distribution of the SST
associated with a DCC for each of the 28 groups.

Figure 1 shows the observed distribution of the SST, $N_{SST}(T)$, the observed distribution of the SST associated with
DCC, $N_{DCC}(T)$, both normalized to unity, for the odd numbered days from 2016. The distribution functions were
evaluated with the Matlab "histc" function with 0.1K wide bins between 280 K and 310 K and a 1K wide
rectangular smoothing kernel.

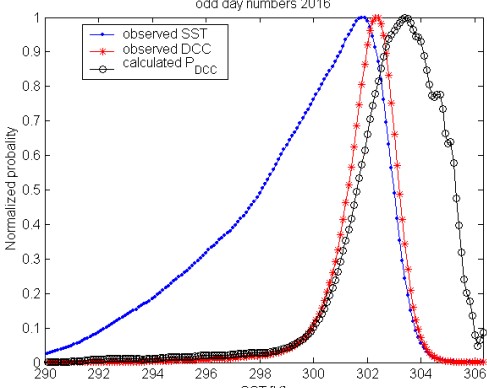

Figure 1. The observed probability of SST and DCC and the calculated probability of the DCC process, normalized
to one.






Assume the surface temperature distribution is $PDF_{SST}(T)$, and we have N samples. By definition of a PDF, for N
samples of the tropical ocean the number of samples at temperature T, is

$$N_{SST}(T) = N*PDF_{SST}(T). \qquad \text{(Eq. 1)}$$


The observed count of DCC in these N samples, $N_{DCC}(T)$, is the product of the temperature distribution of the SST
with the probability of a temperature dependent DCC process, $P_{DCC}(T)$. $P_{DCC}(T)$ is the probability that a DCC
process is associated with a surface temperature T.

For $N_{SST}(T)$ samples of the tropical oceans at temperature T

$\qquad N_{DCC}(T)=N_{SST}(T) * P_{DCC}(T)$ $\qquad\qquad\qquad\qquad\qquad\qquad\qquad\qquad\qquad\qquad$ (Eq.2)

Then

$\qquad P_{DCC}(T) =N_{DCC}(T)/N_{SST}(T)$, for all $N_{SST}(T)>>0$. $\qquad\qquad\qquad\qquad\qquad$ (Eq. 3)

The calculated $P_{DCC}(T)$ (normalized to unity) is shown in Figure 1 overlaid on $N_{SST}(T)$ and $N_{DCC}(T)$. $P_{DCC}(T)$ rises
steeply for temperatures larger than 300K and reaches a peak near 303K, then drops steeply to zero near 306K. We
define the start and end of the DCC process as the temperature where $P_{DCC}(T)$ rises above, then drops below is ½
peak value.

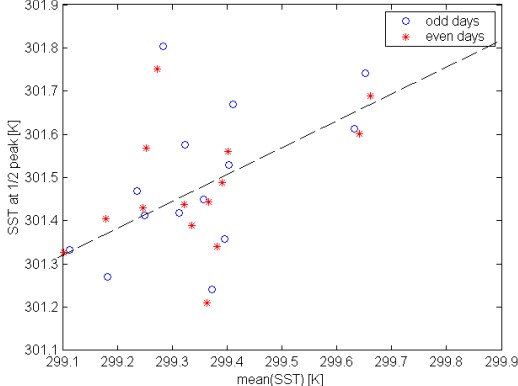

Figure 2. The onset of the DCC process as function of mean temperature of the tropical ocean.
The onset and the end of the DCC formation process was calculated for each of the 28 data groups. The result is
shown in Figure 2 as function of the mean tropical SST. Figure 2 shows that the onset of the DCC process increases
with the mean temperature of the tropical ocean. The analysis of all 28 data groups results in a slope of +0.50±0.18
K/K with warmer SST. This is the convective adjustment of the DCC process. The data from the even and odd days





are statistically independent observations of the same DCC process. For the odd days we find a slope of 0.60 K/K,
for the even days the slope is 0.41 K/K. The difference of 0.19 K/K is consistent with the 0.18 K/K one sigma
uncertainty derived from all data. Within the error bars these results are insensitive to data processing details
(Appendix B).

The end of the DCC process (1/2 peak down probability) is at a temperature of about 304K. We detected no credible
temperature sensitivity of the end of the DCC process. In the following calculations we shift the start and end of the
DCC process at the same rate. The high temperature cut-off of the DCC process is no sooner than 306K. The data
include very few SST or DCC at temperatures warmer than 306K. The 99.9 percentile of all SST and the SST
associated with DCC was 306K.

2.2. Model Data
Precipitation and temperature data from 36 state-of-the-art climate model simulations from the fifth Phase of the
Coupled Model Intercomparison Project (CMIP5) (Taylor et al., 2012) were used for our analysis of the "current
climate" and the "future climate". Here "current climate" refers to mean historical data from 1976-2005 and "future
climate" refers to mean projection from RCP8.5 for the 2070-2100 period. The model output for the tropical ocean
SST and rain rate were averaged in 2.5 degree bins of the tropical oceans. Figure 3 shows the Probability Density
Function (PDF) of the SST, $PDF_{SST}(T)$ for the current SST (blue) and the future SST (red) using 0.5K wide bins.
Except for a shift of about 3K, $PDF_{SST}$ for the current and future temperatures are essentially identical. We conclude
that we can approximate the PDF of a warmer ocean by uniformly shifting the $PDF_{SST}$ deduced from the current
climate.

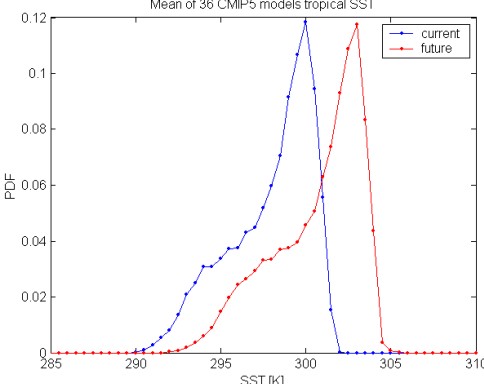

Figure 3. $PDF_{SST}(T)$ for the current SST (blue) overlaid on the PDF of the future SST (red).
**3. Discussion**
3.1. Convective adjustment of the onset of the DCC process.
There are physical reasons to expect that the threshold for the onset of processes related to deep convection will shift
with the temperature and the vertical profiles controlling convective conditional instability (e.g. Williams,





Pierrehumbert and Huber 2009, Neelin 2014). The numerical value of this adjustment depends on the process.
Johnson and Xie (2010), used a mix of satellite and rain gauge data from between 1980 and 2010 to show significant
co-variability between the mean tropical SST and the temperature of the onset of the rain process using a 2 mm/day
rain threshold. The same physical reasoning is imbedded in the climate models. Johnson and Xie (2010) reported
that the theoretically predicted convective adjustment of the Moist Adiabatic Lapse Rate (MALR), 0.43 K/K, ranges
from 0.42 to 0.48 K/K in the CMIP3 models. The temperature of the onset of the rain process for rain rates between
2 and 7 mm/day shifts at the rate of 0.98 K/K (range of 0.86 K/K to 1.05K/K) for the 36 CIMP5 models (Appendix
C). The DCC process is related to very heavy rain, covering 0.6% of the area of the ocean, while the rain process in
the CMIP5 models starts with more moderate rain in a much larger area: The 5 mm/day rain rate covers 23% of the
ocean. The observed convective adjustment of the DCC process from our 14 year data record is close to that of the
MALR in the CMIP3 models and considerably less than that of the rain process in the CMIP5 models.

**3.2. DCC frequency in a warmer climate**
For the calculation of the change in the frequency of DCC we used the $PDF_{SST}$ and $P_{DCC}$ calculated from the mean
the annual values deduced for the 2003-1016 period. In order to account for the effect of the uncertainty in the
convective adjustmeant we used 0.4 K/K, 0.6 K/K and 0.8 K/K for the DCC convective adjustement, referred as as
three scenarios. Figure 4 shows the results from the present to present+4K of warming with 0.1K steps. In the three
scenarios the frequency of the DCC increases 27%, 19% and 9% for the first 1K of global warming (T+1K), and
reaches a peak of 53%, 43% and 23% higher than the present frequency at T+2.7 K. In the current climate 0.6% of
the area of the tropical oceans is accociated with DCC. This increases to between 0.74 to 0.9 percent of the area by
the end of the century, close to a 50% increase.

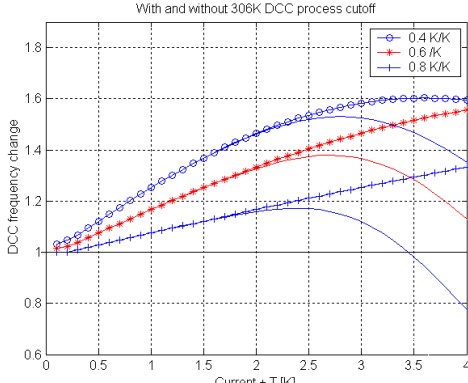

Figure 4. Change in the DCC frequency with warming ocean.

It has been argued (e.g. Hartman and Michelson 1993) that at a temperature warmer than about 303K the heat loss
due to evaporation starts to exceed the solar incident energy. This decreases the probability of the DCC process.
This argument is consistent with the observed peak and subsequent steep roll-off of $P_{DCC}$ seen in Figure 1. Based on
our data we know that the high temperature cut-off of the $P_{DCC}$ in the current climate is no sooner than 306K. The
thin color-coded lines in Figure 4 show the roll-off of the DCC frequency if we assume a 306K high temperature





cut-off of the DCC process. The impact of the 306K cutoff start to become noticeable as a decrease in the DCC
frequence no earlier than T+2.5K. Our results are not significantly changed by using a 205K or 215K threshold for
the definition of DCC.

As the frequency of DCC increases, the contribution of the DCC rain rate to the total rain rate should increases.
However, in order to estimate the magnitude of this effect we need to make a number of uncertain assumptions: 1)
Assume that that the rain rate associated with DCC in the current climate, 3mm/hr, does not change in a future
climate. 2) Assume that the increase in the rain rate is a linear function of the increased number of DCC. 3) Assume
that the resulting increase in the rain rate is not compensated for by a decrease in the rain rate for less cold clouds.
The need for these assumption makes a quantitative estimate of the change in the total rain rate related to the
increase in the DCC frequency uncertain.

**Summary**

We derive the probality of the DCC process as function of the Sea Surface Temperature (SST) associated with DCC
identified with AIRS between the years 2003 and 2016 for the tropical oceans (30S-30N). During this time the
annual mean value of the SST varied by almost 1K, producing measureable shifts in the temperature of the onset of
the DCC process. We find that the temperature of the onset of the DCC formation process shifts at the rate of about
+0.50 K per K of the warming of the mean tropical ocean temperature. We use the probability of the DCC process to
predict the change in the frequency of DCC in a future climate, based on the temperature distribution of the tropical
oceans predicted by the CMIP5 models. The average of the 36 CMIP5 models (RCP 8.5 scenario) predicts 2.7 K of
warming of the mean SST of the tropical oceans by the end of this century. As a result the percent of the area of the
tropical oceans associated with DCC, in the current climate 0.6 percent, increases to 0.9% by the end of the century,
close to a 50% increase.

**Acknowledgements**

The research described in this paper was carried out at the Jet Propulsion Laboratory, California Institute of
Technology, under a contract with the National Aeronautics and Space Administration. We are grateful for the
unwavering support of Dr. Ramesh Kakar of NASA Headquarters. The daily collection of AIRS DCC, merged with
the NOAA SST, is found in the AIRS Calibration Data Subset (ACDS), available freely from the Goddard Earth
Sciences Data and Information Services Center https://disc.gsfc.nasa.gov/AIRS. The AMSRe data are freely
available from ftp://ftp.remss.com/amsre/bmaps_v07/. The CMIP5 RCP 8.5 data can be downloaded from cmip-
pcmdi.llnl.gov/cmip5/data_portal.html







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

**Appendix.**

**A. DCC and Rain rate**

For the determination of the correlation between DCC and rain rate we used AIRS from every 24[th] day (135 days)
between September 2002 and September 2011. The AMSRe data are used to get the rain rate associated with the
positions identified as DCC using the AIRS data. AIRS is +/- 49 degree cross-track scanning with a 1785 km swath
width and 15 km average footprint size. AMSRe scans a cone with a 1450 km swath width and we used the 0.25
degree gridded product. The area covered by AMSRe is covered about two minutes later by AIRS. AMSRe on EOS
Aqua operated from August 2002 through September 2011. The AMSRe data are freely available from
ftp://ftp.remss.com/amsre/bmaps_v07/ with an overview in http://www.remss.com/missions/amsr.

The mean rain rate of 0.11 mm/hr was derived from random sampling the AMSRe data in the AIRS swath. The
fraction of the mean rain rate associated with a threshold is the fraction of the tropical ocean times the mean rain rate
associated with the threshold divided by 0.11. For these data we identified 1,739,919 cloud tops colder than 225K in
the tropical oceans. Of these 483,906 satisfied the bt900 <210K condition for DCC. If we define DCC as colder than





205K only 255,780 DCC are identified. In Table 1 we summarize average rain rates associated with cold cloud tops
selected with 205K, 210K, 215K and 225K thresholds.

| Threshold | 205K | 210K | 215K | 225K |
|---|---|---|---|---|
| Fraction of the tropical oceans associated with the cold cloud | 0.0034 | 0.0064 | 0.0106 | 0.0228 |
| Average rain rate [mm/hr] | 3.61 | 3.06 | 2.61 | 1.89 |
| False alarm rate | 12% | 15% | 18% | 25% |
| Fraction of total rain | 11% | 17% | 26% | 40% |
| $PDF_{DCC}$ ½ up temperature | 301.8 | 301.6 | 301.5 | 301.3 |

Table 1 summarizes results for cold cloud tops selected with 205K, 210K, 215K and 225K thresholds.
The temporal persistence of heavy rain is much shorter than the persistence of associated cold cloud tops. This
temporal mismatch creates the potential for false alarm, i.e. a cold cloud not associated with rain in the AMSRe
record. We define the false alarm fraction as the count of cold clouds with less than 0.1mm/hr of rain divided by the
number of cold clouds associated with each threshold. The false alarm rate for cloud tops warmer than 210K rises
rapidly. The meaning of $PDF_{DCC}$ ½ up will be discussed in Appendix B.

The small fraction of the tropical ocean associated with DCC colder than 210K, 0.6% of the area of the tropical
oceans, puts DCC in the class of extreme objects. According to AMSRe, 0.006 of the area of the tropical ocean is
associated with more than 4 mm/hour of rain. Since the average rain rate associated with DCC is only 3 mm/hr, a
considerable fraction of heavy rain is associated with cloud tops not identified as DCC.

**B. The onset, end and cut-off of the DCC process.**

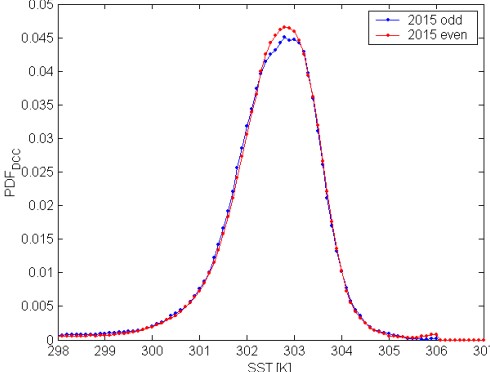


Figure A1. Determination of the $PDF_{DCC}$ from the even and odd day samples of the same year.

Figure A1 shows the $PDF_{DCC}$ for a 210K threshold determined from the even and the odd days in 2015. For the
calculation of N_SST(T) and N_DCC(T) we used the Matlab histc function with a 0.2K wide grid uniformly spaced



between 280 and 310K, and the result was smoothed with a 5 point running mean.We define the onset of the DCC
process as the temperature where the probability reach ½ of its peak.

The temperature of the ½ peak of the PDF, the "onset" of the PDF process", is determined by linear interpolation
between the two points closest to the ½ peak point. Figure 2 in the text shows the 28 calculated values of the
temperature of the onset of the DCC process and associated tropical ocean mean temperatures for the even and odd
day numbers in our data set. The slope, calculated  using the Matlab linear least squares fit, is 0.50 K/K with a slope
uncertainty of 0.18 K/K. The even and odd days independently sample the same DCC process. For the odd days we
find a slope of +0.60 K/K, for the even days the slope is 0.41 K/K. The difference of 0.19 K/K is consistent with the
0.18 K/K one sigma uncertainty.

Alternative numerical methods have been tested on the data, i.e. using the ksdensity function from the Matlab
toolbox for the derivation of N_SST(T) and N_DCC(T). The ksdensity and histc methods produce statistically
consistent slope and slope uncertainty estimates. In all numerical experiments the convective adjustment of the DCC
process is more than two sigma less than 1K/K.

We defined the onset of the DCC process as the temperature where the probability reach ½ of its peak with
increasing temperature. This is intuitively reasonable. If we use 1/3 peak for this definition, the temperature of the
onset of the DCC process shifts slighly colder for all cases, but the slope of the onset as function of the mean SST is
consistent with the slope calculated using the ½ peak.

The accuracy of the determination of the end of the DCC process is limited by the noise amplification inherent in the
deconvolution of $N_{SST.DCC}$ and $N_{SST}$. Analysis of 14 years of data results in a mean value of 303.7K (range 303.1 to
304.5K) with no credible sensitivity to a change in the SST, +0.57±0.75 K/K. Based on our data the cutoff of the
DCC process is no sooner than 306 K. The 99.9 percentile of all SST and the SST associated with DCC was 306K.
The 99.99 percentile was 307K, but at these high temperatures the RTGSST is seldom validated.

Is there a high temperature limit to the DCC process?  In Figure A1 the probability of the observing a DCC process
goes to zero at about 306K. For the 483,906 cold clouds with bt900 <210K there are 168 DCC associated with SST
warmer than 305K. While there are 5850 SST>306.1K in the data covered by the AIRS swath, none are associated
with DCC. This suggests that, at least in the current climate, the DCC process stops no sooner than at 306K.

**C. Convective adjustment of the rain process in CMIP5 models.**

The 36 CMIP5 models give the surface temperature and the rain rate in units of mm/day on a 2.5 degree grid for the
current climate and the future climate. The blue trace in Figure A2 shows the mean rain rate in 1K wide bins of the
surface temperature for the current climate in CMIP5 #16 model. The rain rate increases rapidly as the surface



temperature increases above 299K. The red trace shows the rain rate for the future climate in CMIP5#16. The entire
pattern is shifted about 3K for the future climate. If we define 5 mm/day as a rain process threshold, the threshold is
exceeded at a surface temperature of 300.31K in the current climate, at 303.2K in the future climate. This shift of the
temperature of the onset of the rain process to a warmer temperature in a warmer climate is the convective
adjustment. A similar pattern is seen in all models for the current and the future climate.

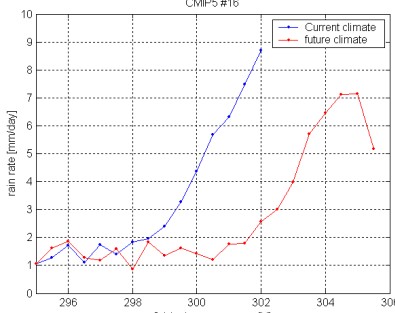

329            Figure A2. Dependence of the rain rate on the on the surface temperature


Between 2 mm/day and 7 mm/day the shift is almost independent of rain rate. The mean convective threshold for the
current climate for a 5 mm/day rain rate is 299.6K (range 298.6K to 301.4K). For the future climate the threshold
shifts to a mean of 302.3K (range 300.6K to 303.8K. At the same time the mean temperature of the tropical oceans
also shifts, depending on the model by between 1.92 K and 3.73K. The ratio of the difference between the
convective threshold for the future and current climate and the difference between the future and current mean
temperature of the tropical ocean is 0.98 K/K on average, with a range of 0.86 K/K to 1.05K/K. 25% of the area of
the current tropical ocean (36 model range 16% to 31%) has a rain rate exceeding 5 mm/day. The roll-off of the
rain-rate at temperatures above 305K is shared by all models.