# Peer review of "Increase in the Frequency of Tropical Deep Convective Clouds with Global Warming1"

_Atmospheric Chemistry and Physics, 2017_

## Referee Comment (RC1) · Anonymous Referee #1 · 30 May 2017

General comments.

This paper concludes that the frequency of DCC will increase in the future climate. This is based on the onset of DCC SST correlated with SST, based on today's inter-annual varability. They used the CMIP climate models to determine the SST change between current and future climate periods. There is also some information provided to the predicted rainfall rate of future DCC. However, the rainfall rate information did not make it into the conclusions or in the abstract. Anytime predictions are made about future climate many assumptions need to be made. The authors primary assumption is to use today's SST inter-annual variability and modeled future SST change to make this prediction. I believe the paper is worthy of publication as one path to get to the frequency of DCC in future climate, the basis of which can be used in future papers on the topic, as more climate change studies and data becomes available.

Specific comments.

I believe the authors have tried to focus this paper more about the assumptions and predictions than about the procedures used. All of the methodology seems to be in appendices and are very succinct and in my opinion lacking some detail. However, more elaboration could help bolster these assumptions, especially section 3.2. I had to read the section several times to see how a DCC SST onset and DCC frequency were tied together.

I found the DCC rainfall rate results in appendix A and C distracted from the intent of the paper. It was not used to determine the increase in frequency of DCC in future climate. It almost seems as an afterthought, while writing the paper. The topic of DCC rainfall rates in future climates is a paper in itself. I also found it odd that none of the rainfall rate information made it into the abstract or conclusions.

This study uses DCC results found over tropical oceans. There is a considerable frequency of DCC found over tropical land. Is the frequency of DCC found over ocean and land correlated in either current or future climates?

Line 51 I am not convinced that the DCC data groups are uncorrelated. I do not agree that the even and odd days from the same year are uncorrelated. Fig. 2 clearly shows that. The even and odd points are located very closely together. Also, the year to year variations could be auto-correlated. In line 271 you state that the even and odd days should provide the same PDF shape.

Fig. 2. This plot uses the inter-annual variability to correlate the onset of convection with SST. The period between 2003 and 2016 were very quiet climate years, except for the 2016 El Nino event. I am sure the El Nino 2016 even and odd day values have a SST ∼299.7 in Fig. 2. Without the year 2016, the slope of the line would be very different. In other words, if the years of 2003 to 2015 were used the conclusions would be different. Would the results look different for 30 years, with volcanic events, and other large El Nino years? For a study that predicts future climate, you have

[Figure]

to assume that today's climate variability can be extrapolated in the future. Today's climate variability is driven by short term climate variability such as the ENSO cycle. This data is the critical information to estimate the increase in DCC frequency. Here are 2 arguments that need to be given some thought.

1) There have been papers that address that inter-annual cloud feedbacks are larger than long-term feedbacks. Zhou, C.,ÂăM. D. Zelinka,ÂăA. E. Dessler, andÂăS. A. KleinÂă(2015),ÂăThe relationship between interannual and long-term cloud feedbacks,ÂăGeophys. Res. Lett.,Âă42,Âă10,463–10,469, doi:10.1002/2015GL066698.

2) Recently studies have pointed out that SST changes is not a very good at predicting inter-annual TOA flux changes, the correlations when using modeled and observed are very weak. Using the 500 hPa tropical temperatures correlations are much greater. (Trenberth, Murphy and Spencer)

Fig. 3. The two SST frequencies of the current and future climates do not look like the SST frequency in Fig. 1. The peak of the current climate is 300K and the future climate is 303K. The peak of fig. 1 using RTGSST SSTs is ~302K. Is it a matter of resolution, 0.25° versus 2.5K?

Fig. C1 Why is there no current climate peak plateau or warm temperature drop off in the rain rate compared with the future climate?

Technical Corrections.

Line 29. I am assuming that the DCC frequency was based on the AIRS footprint (12 km) resolution. DCC smaller than 12-km are not counted. How well does this method separate the anvil or core blow off, which will have a constant temperature, from the convective core? What gives you the confidence that only the DCC cores are identified in the DCC frequency study.

Line 47. I appreciate the author stating the data used in the acknowledgements. Still information is lacking what product and version was used for this analysis. Here it

just states that DCC identified AIRS is collocated with RTGSST (is there a version number?), which is on a 0.25° grid. Was the AIRS (12 km) convolved into the 0.25° resolution data? Or was the RTGSST convolved into the AIRS footprint resolution? What product of AIRS was used and what version of AIRS?

Line 47. I assume that a BT threshold temperature of 210K was used, since that was stated in line 31. Could that threshold be stated again in line 47. The acknowledgement then states this data is publicly available as a product.

Line 55 add "all-sky tropical ocean" in between observed and distribution of the SST for clarity.

Line 248 Can the AMSR-E product name and version number be stated in the text. This is also a 0.25° gridded product as was the RTGSST. Again, was the AIRS footprint data also mapped into 0.25° gridded regions?

Line 270 Fig. A1. I believe should be Fig. 1

---

## Author Comment (AC1) · 6 Jun 2017

Author's response to anonymous referee #1.

Based on the quoted line numbering and Figure numbers referee#1 read the originally submitted version of the manuscript. In response to the editors request a revised manuscript was submitted (and is on the discussion page).It addresses some of the questions raised by the reviewers.

====================================

General comments.

This paper concludes that the frequency of DCC will increase in the future climate.

[Figure]

This is based on the onset of DCC SST correlated with SST, based on today's inter-annual variability. They used the CMIP climate models to determine the SST change between current and future climate periods. There is also some information provided to the predicted rainfall rate of future DCC. However, the rainfall rate information did not make it into the conclusions or in the abstract. Anytime predictions are made about future climate many assumptions need to be made. The author's primary assumption is to use today's SST inter-annual variability and modeled future SST change to make this prediction.

I believe the paper is worthy of publication as one path to get to the frequency of DCC in future climate, the basis of which can be used in future papers on the topic, as more climate change studies and data becomes available.

===================================

Specific comments.

I believe the authors have tried to focus this paper more about the assumptions and predictions than about the procedures used. All of the methodology seems to be in appendices and are very succinct and in my opinion lacking some detail. However, more elaboration could help bolster these assumptions, especially section 3.2. I had to read the section several times to see how a DCC SST onset and DCC frequency were tied together.

Response:

We have added some clarity to section 3.2 by starting it with the following sentences: "As the PDF of the SST, PDFSST, shifts to warmer temperatures, the PDF of the DCC process, PDCC, shifts at a slower rate. This causes the number of DCC (calculated from Eq.2) to increase relative to the current climate. For the calculation of the change in the frequency of DCC we used the PDFSST and PDCC calculated from the mean the annual values deduced for the 2003-1016 period, both appropriately shifted to account

for the warmer climate."

====================================

I found the DCC rainfall rate results in appendix A and C distracted from the intent of the paper. It was not used to determine the increase in frequency of DCC in future climate. It almost seems as an afterthought, while writing the paper.

Response:

The rainfall rate in Appendix A is included to support the argument that the majority of the DCC detected with the 210K threshold are associated with the heavy rain rates associated with the convective cores. Only a small fraction (15%) are false alarms (Table 1).

Appendix C with rain rates in the CIP5 models and references to it in the main body of the manuscript have been removed.

====================================

The topic of DCC rainfall rates in future climates is a paper in itself. I also found it odd that none of the rainfall rate information made it into the abstract or conclusions.

Response:

We agree. As stated in the discussion, as the frequency of DCC increases, the contribution of the rain associated with DCC to the total rain rate may also increase. However, an estimate the magnitude of this effect requires a number of uncertain assumption, which make quantitative estimates of the expected change in the total rain rate uncertain.

====================================

This study uses DCC results found over tropical oceans. There is a considerable frequency of DCC found over tropical land. Is the frequency of DCC found over ocean

and land correlated in either current or future climates?

Response:

In "Aumann H. H. and A. Ruzmaikin, 2013: Frequency of deep convective clouds from ten years of AIRS data, ACP 13, 1-39, doi:10.5194/acp-13-1-2013" we discussed the DCC frequency over tropical land and ocean. It showed that the ocean/land DCC frequencies between 2002 and 2012 were anti-correlated. The problem is that the interpretation of trends in a time series in the presence of large natural variability can't credibly be extended to a future climate.

====================================

Line 51 I am not convinced that the DCC data groups are uncorrelated. I do not agree that the even and odd days from the same year are uncorrelated.

Response:

The mean SST from consecutive days at any location is highly correlated. We also agree that measurements of the onset of the DCC process for each year is correlated, since we are measuring the same process twice for the same year. However, the measurements of the process is noisy: 1) While on average 4000 DCC are identified in the day+night overpasses of the tropical oceans on any one day, the standard deviation is about 1000. The next day is likely to find only 2000 or as many as 6000 DCC located somewhere else in the tropical oceans.

2) The time/position latency of a DCC relatively short. The probability that a DCC identified at the 1:30 AM morning overpass is identified with another DCC within 50 km 12 hours later in the 1:30 PM overpass is only 20%. The probability to find a DCC within 50 km one day later is very small.

These two reasons add noise to the calculation of the onset of convection from the even and odd days of each year, which de-correlates the measurements. As pointed out in the manuscript, this is the reason that the separate analysis of the even and odd data

results in a slope and slope uncertainty which has the characteristics of independent samples drawn from the same distribution: For the odd days we find a slope of 0.60 K/K, for the even days the slope is 0.41 K/K. The difference of 0.19 K/K is consistent with the 0.18 K/K one sigma uncertainty derived from all data.

=====================================

Also, the year to year variations could be auto-correlated. In line 271 you state that the even and odd days should provide the same PDF shape.

Response:

We were not able to find the quoted text since the manuscript line numbers start with each page. The manuscript states: "The data from the even and odd days are statistically independent observations of the same DCC process."

=====================================

Fig. 2. This plot uses the inter-annual variability to correlate the onset of convection with SST. The period between 2003 and 2016 were very quiet climate years, except for the 2016 El Nino event. I am sure the El Nino 2016 even and odd day values have an SST=299.7 Kin Fig. 2. Without the year 2016, the slope of the line would be very different. In other words, if the years of 2003 to 2015 were used the conclusions would be different.

Response:

2016 was indeed the warmest. 2015 was the next warmest. 2008 and 2011 were the two coldest years in the 14 year data set.

The slope without the even and odd days from 2016 is +0.46 with a probable error of 0.22. The mean value of the slope did not change within the probable error, but, as expected, the slope uncertainty increased. For our 14 year data set the slope of the linear regression is dominated by the coldest years (2008 and 2011) and the warmest

years (2015 and 2016). Ignoring all other years the slope is 0.57 K/K, essentially the same as when using all 2003-2016 data.

==================================

Would the results look different for 30 years, with volcanic events, and other large El Nino years?

Response:

We expect the analysis of extreme years to produce the same shift (within the error bars) in the onset of the DCC process as observed for the 2003 to 2016 data.

==================================

For a study that predicts future climate, you have to assume that today's climate variability can be extrapolated in the future. Today's climate variability is driven by short term climate variability such as the ENSO cycle. This data is the critical information to estimate the increase in DCC frequency.

We agree. We have to assume that extremes due to natural variability of the current climate provided by ENSO events capture information useful for predicting extremes in a future climate. The soundness of our assumption is supported by the fact that the onset of the deep convective process is expected on thermodynamic grounds to change +0.43 K/K in a warmer climate. This is consistent with the +0.5K/K convective adjustment of the onset of the DCC process derived in our paper based on the extremes of natural variability of the current climate.

==================================

Here are 2 arguments that need to be given some thought.

1) There have been papers that address that inter-annual cloud feedbacks are larger than long-term feedbacks. Zhou, C., M. D. Zelinka, A. E. Dessler, and S. A. Klein (2015),The relationship between interannual and long-term cloud feed- backs, Geophys. Res. Lett. 42, 10,463–10,469, doi:10.1002/2015GL066698.

2) Recently studies have pointed out that SST changes is not a very good at predicting inter-annual TOA flux changes, the correlations when using modeled and observed are very weak. Using the 500 hPa tropical temperatures correlations are much greater. (Trenberth, Murphy and Spencer).

Response:

The Zhou et al., which was based on CMIP modeling, discusses the radiative cloud feedbacks. It supports our main finding of DCC frequency increase with the rise of SST. As stated in that paper "On both timescales (interannual and long-term), the increase in high cloud fraction and decrease in middle- and low-level cloud fraction are significantly larger in models with large long-term ( timescales)".

We are not familiar with the Trendberth et al. paper. The TOA flux presumable comes from CERES. AIRS and CERES are on the EOS Aqua spacecraft. DCC identified with AIRS on a 0.2 degree lat/lon scale are heavily diluted in the CERES OLR and RSW products presented on a one degree grid. The changes in the 500 hPa tropical temperatures correlate with the convective adjustment of the Moist Adiabatic Lapse Rate (MALR), which matches the observed adjustment of the DCC process.

====================================

Fig. 3. The two SST frequencies of the current and future climates do not look like the SST frequency in Fig. 1. The peak of the current climate is 300K and the future climate is 303K. The peak of fig. 1 using RTGSST SSTs is 302K. Is it a matter of resolution, 0.5o versus 2.5 degree?

Response:

The 36 CMIP5 models all have different bin sizes, two have bins as small as 1.1 degree, some have bins larger than 2.5 degrees. Figure 3 was created by mapping the 36 models to a common 2.5 degree grid. The main point of Figure 3 is to illustrate the

nearly linear shift in the PDF of the tropical SST along the horizontal axis from the current to the future climate. The RTGSST, which was used to create PDFSST, is based on a 0.5 degree grid.

====================================

Fig. C1 Why is there no current climate peak plateau or warm temperature drop off in the rain rate compared with the future climate?

Response:

You are referring to Appendix C Figure A2. This appendix has been deleted. As the reviewer suggested, it is a distraction for the DCC focus of the paper. In addition, we are concerned that interpolating rain rates to a common grid and averaging the result could produce misleading results.

====================================

Technical Corrections.

Line 29. I am assuming that the DCC frequency was based on the AIRS footprint (12 km) resolution. DCC smaller than 12-km are not counted. How well does this method separate the anvil or core blow off, which will have a constant temperature, from the convective core? What gives you the confidence that only the DCC cores are identified in the DCC frequency study.

Response:

This issue is addressed in Appendix A. The temporal persistence of heavy rain (associated with the convective core) is much shorter than the persistence of cold anvil or core cirrus blow off. With the DCC identified with the 210K threshold, only 15% are not associated with intense rainfall as reported by AMSRe 5 minutes earlier. This gives us confidence that the DCC used in our study are associated with deep convective cores.

====================================

[Figure]

Line 47. I appreciate the author stating the data used in the acknowledgements. Still information is lacking what product and version was used for this analysis.

Response:

The Acknowledgements include the link to the AIRS Calibration Data Subset Version 6.0 at the htpps//disc.gsfc.nasa.gov/AIRS. This subset uses the AIRS Version 5.0 calibrated radiances. Here it just states that DCC identified AIRS is collocated with RTGSST. Is there a version number?), which is on a 0.25 degree grid.

Response:

Actually the RTGSST is available on a 0.5 degree grid. This was incorrectly stated in the manuscript. The Acknowledement states that source as ftp://polar.ncep.noaa.gov/pub/history/sst/rtg_low_res). We did not use the more recent rtg_high-res version.

====================================

Was the AIRS (12 km) convolved into the 0.25 degree resolution data? Or was the RTGSST convolved into the AIRS footprint resolution?

Response:

We used the RTGSST from the gridpoint closest to the position of the DCC identified with the AIRS data.

====================================

What product of AIRS was used and what version of AIRS?

I assume that a BT threshold temperature of 210K was used, since that was stated in line 31. Could that threshold be stated again in line 47. The acknowledgement then states this data is publicly available as a product.

Response:

[Figure]

We used the AIRS Calibration Data Subset (ACDS) Version 6.0 at https//disc.gsfc.nasa.gov/AIRS. This subset uses the AIRS Version 5.0 calibrated radiances (AIRS L1b) product. A subset of the ACDS includes all footprints between latitudes 50S and 50N where bt900<225K, i.e. cold cloud features. This allows to user re-filter for clouds which include a convective core. We used bt900<210K and 30S-30N ocean. and the brightness temperatures from AIRS channel #759 at 900.6 cm-1. The radiances use the AIRS prelaunch calibration. It is anticipated that a future Version 6 will add some diagnostic fields, but no changes to the radiances appear to be justified at this time (June 2017).

=====================================

Line 55 add "all-sky tropical ocean" in between observed and distribution of the SST for clarity.

Response:

Good point. Done

=====================================

Line 248 Can the AMSR-E product name and version number be stated in the text.

Response:

As stated in the Acknowledgements, The AMSRe data are freely available from ftp://ftp.remss.com/amsre/bmaps_v07/. The overview is in http://www.remss.com/missions/amsr. This is a daily 0.25 degree gridded product, which separates the ascending and descending orbits to allow a perfect match to the AIRS spatial coverage about two minute later.

=====================================

Again, was the AIRS footprint data also mapped into 0.25 degree gridded regions?

Response:

We used the AMSRe rain rate from the gridpoint closest to the position of the DCC identified with the AIRS data

==================================

Line 270 Fig. A1. I believe should be Fig. 1

Response:

Figure A1 (in Appendix A) shows the PDF of the DCC observations for the even and odd days from 2015 as a function of local SST. It shows that the even and odd days are not exactly the same. Figure 1 (red curve) shows the PDF of the DCC observations for 2016, and the random samples SST distribution (blue) and the derived PDF of the DCC process.

---

## Referee Comment (RC2) · Anonymous Referee #2 · 29 Jul 2017

Review of 'Increase in the frequency of tropical deep convective clouds with global warming' by H. H. Aumann and co-authors for consideration in ACP.

In this manuscript the authors take observations of tropical deep convective clouds (DCC), roughly reaching the tropopause, and relate their occurrence to the underlying SSTs. Using data from the period 2003-2016 they find that the likelihood distribution of DCC shifts to warmer SSTs slower than the tropical mean SST at a rate of about 0.5 K/K. This means that in the warmer years a larger area is covered with DCC. The authors extrapolate this to the future using SST distribution change projected using climate models, to yield the main result of the study that there might be a 50 percent increase in the occurrence of DCC.

The result is challenging both because models do not seem to reproduce the behavior,

and because there seems to be little theoretical grounds for this. All other things equal, in a warmer climate the largest storms are expected to increase their precipitation with Clausius-Clapeyron, i.e. about 7 percent per Kelvin. If we assume DCC are such storms, then their frequency of occurrence would have to decrease slightly because to total precipitation is bound by radiative cooling which gives only 2-3 percent per Kelvin increase. There could, of course, be a shift from other types of precipitation to DCC, but I doubt it could change this fast, and would argue that there are simpler explanations for the statistical result.

The first is purely methodological. In figure 2 regression is used to fit a line to the data points, the result has a slope of about 0.5 K/K. Linear regression is usually used when one variable is much less noisy than the other, placing the less noisy variable on the x-axis. Judging from the even/odd days pairs there does seem to be a bit more noise in the "SST at 1/2 peak" than the tropical mean SST, but it is by far not dominant and it further cannot explain deviations from the line. In this case it is better to use a total least squares method (sometimes orthogonal linear regression). This will probably yield a slope that is indistinguishable from 1 K/K. Probably Matlab does not have this capability out of the box, but there is help to be found on the internet. The authors might also simply try to invert the order of regression variables, and then invert the slope back again, which will probably yield slopes » 1 K/K.

The second is conceptual. In the same figure 2 it is obvious that most of the signal contributing to the slope of about 0.5 K/K comes from the two warm years with tropical mean SST of about 299.6 K. These, I assume, are 2015 and 2016 which were El Nino years. The distribution of SST is quite different in years with El Nino compared to normal years, and it is quite different from that of global warming as predicted by climate models, in that the warmest SSTs from the warm pool region are spread out eastward such that deep convection can occur over a larger area. Perhaps then it is not surprising that the 1/2 peak DCC distribution is not moving to warmer SSTs as fast as the tropical mean, since the warmest SSTs are not warming substantially during El

Nino.

Overall, the physical concepts, the linear statistical assumption and data underlying this study and used to project into the future leaves much to be desired, and I fear the found signal is pure artefact. I can therefore not recommend publication of this study. I therefore also refrain from going into details with the weak presentation, language and problematic referencing to the literature.

---

## Author Comment (AC2) · 2 Aug 2017

We thank reviewer#2 for several penetrating comments.

1) . . . If we assume DCC are such storms, then their frequency of occurrence should decrease slightly. . ..

Response: The Clausius Clapeyron argument applies to mean values, but we are not so sure that conclusions for the mean apply to the extremes, the DCC, used in our study. The DCC are not typical large storms, but the severest form of thunder storm, where the cloud tops penetrate the tropopause in the area of the AIRS field of view (15x15 km) and larger. DCC are by far not the dominant source of the mean tropical rain fall. DCC cover only 0.6% of the tropical oceans (on any one day). They are

associated with 3 mm/hr of rain (according AMSRe). Since the mean rain rate for the tropical ocean is 0.1mm/hr, they contribute 0.006*3/0.1= 18% to the total tropical ocean rainfall. If the frequency of the DCC goes up by 50% in a warmer climate and if the associated rain rate is still 3mm/hr, then the contribution of the DCC to the total rain fall is 27%. This still is not the dominant source of mean rain fall, but not insignificant. There may be compensating effect, e.g. less shallow convection, in a warmer climate. Resolving this is a challenge for models. We hope that by getting this paper into the literature, we start the process.

2) The result (0.5 K/K) is challenging because models do not produce the reported result and because there seems to be little theoretical grounds for this.

Response: DCC are very difficult to model, even on the 1/4 degree grid used by ECMWF. There should be no expectation that evidence for DCCs should be seen in climate models due to their much lower spatial resolution than AIRS. However, there are indirect indicators related to vertical stability which do not follow the surface warming by 1K/K. The Moist Adiabatic Lapse Rate (MALR) at 300 hPa is a measure of vertical stability. As stated in the manuscript, the paper by Johnson and Xie (2010, Figure 2a in the paper and associated discussion) finds that MALR in the CMIP3 models has a convective adjustment of between 0.42-0.48 K/K. They claim that 0.43K/K is expected on theoretical grounds. Intuitively there should be a relationship between MALR and DCC, but it have not been established.

3) Methodology: Linear regression is usually used when one variable is much less noisy than the other and the less noisy variable is placed on the x-axis.

Response: Figure 2 shows that the mean SST (x-axis) is by far less "noisy" than the SST at $\frac{1}{2}$ peak. The mean SST on the x-axis are not random variables. The fact that there is any difference in the mean SST between the odd and even days is due to the fact that the SST for the tropical ocean is random sampled by only about 7,000 points per day.

4) Methodology: A more sophisticated software would probably yield a slope closer to 1 K/K.

Response: We challenged the 0.5K/K result by using different algorithm, including dividing the data into odd and even days, rejecting the warmest two years (more on this further down) and having one of the coauthors repeat the analysis from scratch. The quoted results used the MATLAB toolbox ("out of the box") because it is a traceable, transparent, simple linear least squares fit to the data. We tried REGRESS and PLSREGRESS from the MATLAB statistics toolbox. However, PLSREGRESS is a PCA type regression, which treats the x and y axis as random variables. This is not appropriate for our data and the result was rejected. (The PLSRERGESS produced a slope of 0.11 K/K with 0.05 K/K slope uncertainty.) Nothing got the slope anywhere near 1K/K.

5) Conceptual: …. It is obvious the most of the signal contributing to the slope of 0.5K/K comes from the two warmest years, the 2015 and 2016 El Nino years…. Perhaps then it is not surprising that the $\frac{1}{2}$ peak of the DCC distribution is not moving to warmer SST as fast as the tropical mean….

Response: The reviewer is correct that 2015 and 2016 are the warmest two years. We interpret "moving to warmer SST as fast as the tropical mean" implies 1 K/K. As a numerical experiment we removed the four data points where the mean tropical SST is warmer than 299.6 K. The slope decreased to 0.19K/K, but with a much larger 0.34K/K slope uncertainty. This is in the opposite direction of that expected by the reviewer. Even the mean plus one sigma is still far from 1 K/K.